# Crop wild relative populations of *Beta vulgaris* allow direct mapping of agronomically important genes

Gina G. Capistrano-Gossmann[1], D. Ries[2], D. Holtgräwe[2], A. Minoche[3,4], T. Kraft[5], S.L.M. Frerichmann[1], T. Rosleff Soerensen[2], J.C. Dohm[6], I. González[7], M. Schilhabel[1], M. Varrelmann[8], H. Tschoep[9], H. Uphoff[5], K. Schütze[10], D. Borchardt[10], O. Toerjek[10], W. Mechelke[10], J.C. Lein[10], A.W. Schechert[11], L. Frese[12], H. Himmelbauer[3,6,7], B. Weisshaar[2] & F.J. Kopisch-Obuch[1,10]

Rapid identification of agronomically important genes is of pivotal interest for crop breeding. One source of such genes are crop wild relative (CWR) populations. Here we used a CWR population of <200 wild beets (*B. vulgaris* ssp. *maritima*), sampled in their natural habitat, to identify the sugar beet (*Beta vulgaris* ssp. *vulgaris*) resistance gene *Rz2* with a modified version of mapping-by-sequencing (MBS). For that, we generated a draft genome sequence of the wild beet. Our results show the importance of preserving CWR *in situ* and demonstrate the great potential of CWR for rapid discovery of causal genes relevant for crop improvement. The candidate gene for *Rz2* was identified by MBS and subsequently corroborated via RNA interference (RNAi). *Rz2* encodes a CC-NB-LRR protein. Access to the DNA sequence of *Rz2* opens the path to improvement of resistance towards rhizomania not only by marker-assisted breeding but also by genome editing.

[1] Plant Breeding Institute, Kiel University, Am Botanischen Garten 1-9, Kiel 24118, Germany. [2] CeBiTec & Faculty of Biology, Bielefeld University, Universitätsstraße 25, Bielefeld 33615, Germany. [3] Max Planck Institute for Molecular Genetics, Ihnestraße 73, Berlin 14195, Germany. [4] Garvan Institute of Medical Research, 384 Victoria Street, Darlinghurst, Sydney NSW 2010, Australia. [5] Syngenta Seeds AB, Box 302, Landskrona 26123, Sweden. [6] Department of Biotechnology, University of Natural Resources and Life Sciences (BOKU), Muthgasse 18, 1190 Vienna, Austria. [7] Centre for Genomic Regulation (CRG), Carrer del Dr. Aiguader 88, Barcelona 08003, Spain. [8] Department of Phytopathology, Institute of Sugar Beet Research (IfZ), Holtenser Landstraße 77, Göttingen 37079, Germany. [9] SESVanderHave N.V., Industriepark, Tienen 3300, Belgium. [10] KWS SAAT SE, Grimsehlstraße 31, Einbeck 37555, Germany. [11] Strube Research GmbH & Co. KG, Hauptstraße 1, Söllingen 38387, Germany. [12] Federal Research Centre for Cultivated Plants (JKI), Erwin-Baur-Str. 27, Quedlinburg 06484, Germany. Correspondence and requests for materials should be addressed to B.W. (email: bernd.weisshaar@uni-bielefeld.de) or to F.J.K.-O. (email: Friedrich.KopischObuch@kws.com).

Crop wild relatives (CWRs) are an invaluable genetic resource for crop improvement. CWR populations collected in their natural habitat not only provide access to agronomically important traits but can also serve to directly identify genes underlying these traits. Gene identification has been done so far mainly in three types of populations: germplasm collections, elite breeding materials, and synthetic populations such as biparental or multi-parent advanced generation inter-crossed populations[1,2]. Compared to those populations, CWR populations have usually undergone many generations of outcrossing and thus display low linkage disequilibrium and high population admixture.

Rhizomania is the most important sugar beet disease next to infection by beet cyst nematode (*Heterodera schachtii*). It is caused by the beet necrotic yellow vein virus (BNYVV) and reduces the yield of sugar beet by up to 80%. The BNYVV virus is transmitted by the soil-borne plasmodiophoromycete *Polymyxa betae*[3]. Starting from the first appearance in Italy in 1952, rhizomania has spread to almost all sugar beet-growing areas of the world[4]. The only way to control this soil-borne disease is through the cultivation of rhizomania-resistant varieties that have been made available since the mid-1980s (ref. 5). Resistance breeding has relied mostly on the resistance genes *Rz1* and *Rz2* as single loci or a combination of both[6,7]. The resistance conferred by *Rz1* has been discovered in sugar beet germplasm, while *Rz2* is derived from the wild beet *Beta vulgaris* ssp. *maritima*. Alleles that provide resistance show a dominant inheritance at both loci, and *Rz2* confers a higher resistance level than *Rz1* (ref. 8). *Rz1* and *Rz2* were mapped at a distance of about 20 cM on chromosome (Chr) 3 of the sugar beet[6,9,10]. A potentially additional rhizomania-resistance locus has been mapped to sugar beet Chr3 (BvChr3) and is most likely identical to *Rz2* as it is derived from the same *B. vulgaris* ssp. *maritima* origin in Denmark[11].

Here we demonstrate the potential of a CWR population for gene-level resolution mapping by identifying the sugar beet (*B. vulgaris* ssp. *vulgaris*) rhizomania-resistance locus *Rz2* in a wild beet population of *B. vulgaris* ssp. *maritima* through a modified version of mapping-by-sequencing (MBS)[12]. The results not only confirm, in yet another case, the potential of CWR for crop improvement and the importance of preserving CWR in their natural habitats but also provide access to an agronomically important gene conferring resistance against rhizomania that encodes a CC-NB-LRR protein.

## Results

**A CWR population to identify a candidate for *Rz2* by MBS**. Aiming at fine mapping and identification of the *Rz2* gene, we sampled seeds from individual plants of the *B. vulgaris* ssp. *maritima* CWR population that is growing along the coast of Kalundborg, Denmark (Fig. 1). This population, referred to as Kalundborg population, is known to segregate for *Rz2* (ref. 6). Given the allogamous flowering biology of this wild beet species, we assumed thorough population admixture and low linkage disequilibrium in the population, which are both features required for high-resolution mapping[13].

We crossed 189 different parental wild beets with a rhizomania-susceptible sugar beet line and used the resulting 189 full sib families for a replicated progeny test. This test for rhizomania resistance consisted of four independent greenhouse assays, each including semiquantitative immuno-detection of BNYVV virions. The families showed segregation for rhizomania resistance and allowed inference of the genotype at the *Rz2* locus of the respective parental wild beets (homozygous resistant (*Rz2/Rz2*), heterozygous (*Rz2/rz2*) or homozygous susceptible (*rz2/rz2*)). The results were explained by assuming a single locus,

showing that *Rz2* is the only rhizomania-resistance locus segregating in the Kalundborg population. Referring back to their original location, resistant genotypes appeared to be randomly distributed along the coastal line (Fig. 1, Supplementary Data 1). For most of the markers, the allele frequencies did not deviate significantly from Hardy–Weinberg equilibrium (HWE; Supplementary Table 1) and thus suggest a random mating population.

For MBS, we selected only parental wild beet individuals with high confidence in phenotype classification and used them to create one bulk of only four homozygous-resistant wild beets and one bulk with eight homozygous-susceptible wild beets. The two bulks were each sequenced using the Illumina technology (Supplementary Table 2). For access to a useful reference sequence for read mapping, the *Rz2* donor accession WB42 (ref. 6,7) was sequenced and assembled to draft quality (Supplementary Table 3). The assembly was concatenated according to the sugar beet reference sequence version 1.2 (ref. 14) to yield pseudochromosomes. RefBeet1.2 covers 567 Mbp of the *B. vulgaris* ssp. *vulgaris* genome that has an estimated haploid genome size of about 730 Mbp in nine chromosomes. The initial assembly, which was designated WB42-v1, was subsequently corrected manually with regard to the positioning of WB42 scaffolds and contigs to BmChr3 to yield the optimized assembly WB42-v2.

MBS analysis[15] with WB42-v2 resulted in the detection of four closely connected intervals within a genomic region of 0.72 Mbp on BmChr3 (Fig. 2a,b, Supplementary Figs 1 and 2), overlapping with 25 almost continuous scaffolds and contigs of WB42-v2. Four smaller intervals spanning altogether about 0.09 Mbp at three loci were found on three other chromosomes (Supplementary Fig. 1, Supplementary Table 4). In the case of the scaffolds and contigs that cause signals outside of BmChr3, even detailed manual inspection did not allow us to deduce a reliable position in the genome sequence. However, mapping with additional markers indicated that these sequences also belong to BmChr3 but are not closely linked to the resistance locus defined by the phenotype (Supplementary Table 5). This excludes these three loci as possible location for *Rz2*. Gene prediction was carried out using the WB42-v2 assembly of *B. vulgaris* ssp. *maritima* together with RNA-Seq read evidence generated from tissue derived from rhizomania-infected plants related to line WB42. A total of 33,922 genes with expression evidence were predicted, including 60 genes in the target region on BmChr3. DNA and deduced peptide sequences of these 60 genes, and of 14 additional genes from the same region without expression evidence, were compared to the NCBI NR protein database and the current *B. vulgaris* annotation BeetSet-2 (ref. 16) (Supplementary Data 2). We detected a candidate for *Rz2* that (i) encodes a CC-NB-LRR protein, (ii) is absent from BeetSet-2 obtained from the susceptible genotype KWS2320 and (iii) was supported by 100% transcript evidence in the WB42-v2 gene prediction. The *Rz2* candidate had been included in the initial *B. vulgaris* gene prediction[14] with the ID *Bv3_jumg* as a truncated gene but omitted from BeetSet-2 due to the absence of detectable transcription in the reference genotype KWS2320. Apart from its truncation in the reference sequence genotype, *Bv3_jumg* was located in a region that displays strong synteny between WB42 and KWS2320. The region was well assembled, co-linear and displayed no indication of read coverage deviation (Supplementary Fig. 3), indicating that no duplicated or paralogous loci from elsewhere in the genome interfered.

**Fine mapping of the *Rz2* candidate to rhizomania resistance**. A total of 33 single-nucleotide polymorphisms (SNPs) segregating in the Kalundborg population, and distributed over the 0.72-Mbp

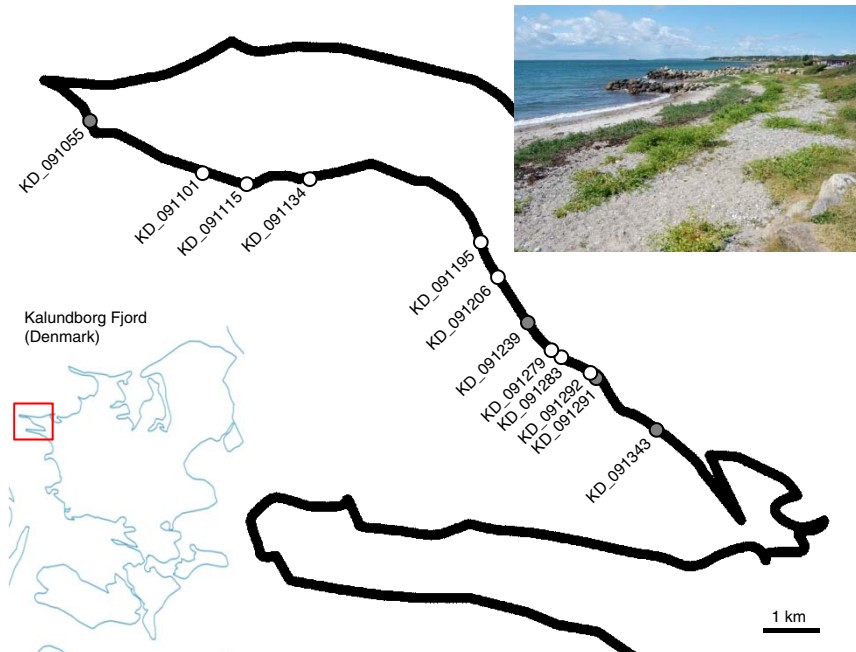

**Figure 1 | A view of the wild beet population of *Beta vulgaris* subspecies *maritima* located along the coastal line of the Kalundborg fjord in Denmark.** The distribution of resistant and susceptible plants selected for MBS, after evaluation for rhizomania resistance, is shown. The exact position of all 189 collected plants used for seed production is presented in Supplementary Data 1. Distribution of susceptible and resistant plants (white and grey circles, respectively) was observed all over the coast.

target region, were converted to molecular markers (see Supplementary Table 6) and used to genotype the complete panel of the 189 parental wild beets (Supplementary Data 1). By association analysis of the marker results with the corresponding resistance phenotype data, 19 markers were significantly associated with resistance in at least two independent rhizomania-resistance tests (Fig. 2c; Supplementary Table 7). The best association was observed for marker CAU3880, which detected an SNP within the gene *Bv3_jumg* that encodes the CC-NB-LRR candidate for *Rz2*. Notably, CAU3880 was also the only marker that displayed association with rhizomania resistance (single marker F-test, $P$ value = 6.1E-4, $r^2 = 0.17$) in an independent second CWR population that was sampled in Brighton, France (see Supplementary Data 3). Therefore, we hypothesized that the CC-NB-LRR gene *Bv3_jumg* containing CAU3880 is the best candidate for *Rz2*.

To validate this hypothesis, the markers with the highest association in the CWR populations were tested using diversity panels of sugar beet inbred lines representing the germplasm of three breeding companies (Supplementary Table 8). Across this association panel, only markers CAU3880, CAU4188 and CAU3882 showed perfect co-segregation with rhizomania resistance. In addition, detailed analysis of the recombination events detected in the Kalundborg CWR population showed that neighbouring genes upstream of *Rz2* were excluded by CAU3881, while genes downstream of *Rz2* were excluded by CAU4188. In the Kalundborg CWR population, the marker CAU4188 also excluded an ankyrin repeat encoding gene that is located downstream of *Rz2* and included in BeetSet-2 (*Bv3_tftt*, see Supplementary Data 2). We concluded that *Rz2* is the only gene that is fully associated with resistance to rhizomania.

**Validation of *Rz2* as the rhizomania-resistance gene by RNAi.** To finally corroborate *Rz2* (*Bv3_jumg*) as the rhizomania-resistance gene, an RNAi experiment was performed using the resistant sugar beet genotype 6921_RR. An RNAi construct was designed to inactivate the intact allele of $Rz2_{KD-R}$, and this construct was transformed into 6921_RR. The transgenic line displays sensitivity to rhizomania to an extent comparable to the susceptible control (Supplementary Fig. 4).

Sequence analysis of the available *Rz2* alleles revealed that alleles associated with susceptibility display defects in the CC-NB-LRR gene *Bv3_jumg*. The susceptible allele from the Kalundborg population ($rz2_{KD-S}$) contained a premature stop codon, and the allele from the susceptible sugar beet reference genotype KWS2320 contained a transposon insertion ($rz2_{KWS2320}$; Fig. 2d,e, Supplementary Table 9). We concluded that an intact allelic version of *Bv3_jumg* represents *Rz2* and that susceptibility to rhizomania is caused by the absence of a functional allele of this gene. This conclusion is further supported by the similarity of *Rz2* and the encoded CC-NB-LRR protein to other identified plant virus-resistance genes of the same type[17].

## Discussion

The CC-NB-LRR family of the disease-resistance genes, which includes *Rz2* on the basis of high amino acid sequence similarity, has been named after the domains that these resistance-conferring receptors typically contain: a nucleotide-binding site (NB) domain, a leucine-rich repeat (LRR), and a coiled-coil (CC) domain. CC-NB-LRR proteins can recognize a wide variety of taxonomically unrelated pathogens, including viruses, bacteria, fungi and even insects[18]. Activation of these genes results in a hypersensitive response characterized by the rapid death of cells in a local region surrounding an infection site, restricting the growth and spread of pathogens to other parts of the plant[19].

The identification of *Rz2* in a CWR population of an allogamous diploid species demonstrates the potential for high-resolution mapping of major genes. The small population size ($N < 200$) drastically reduces the cost for phenotyping and marker assays. In comparison, when working with synthetic populations, 8,283

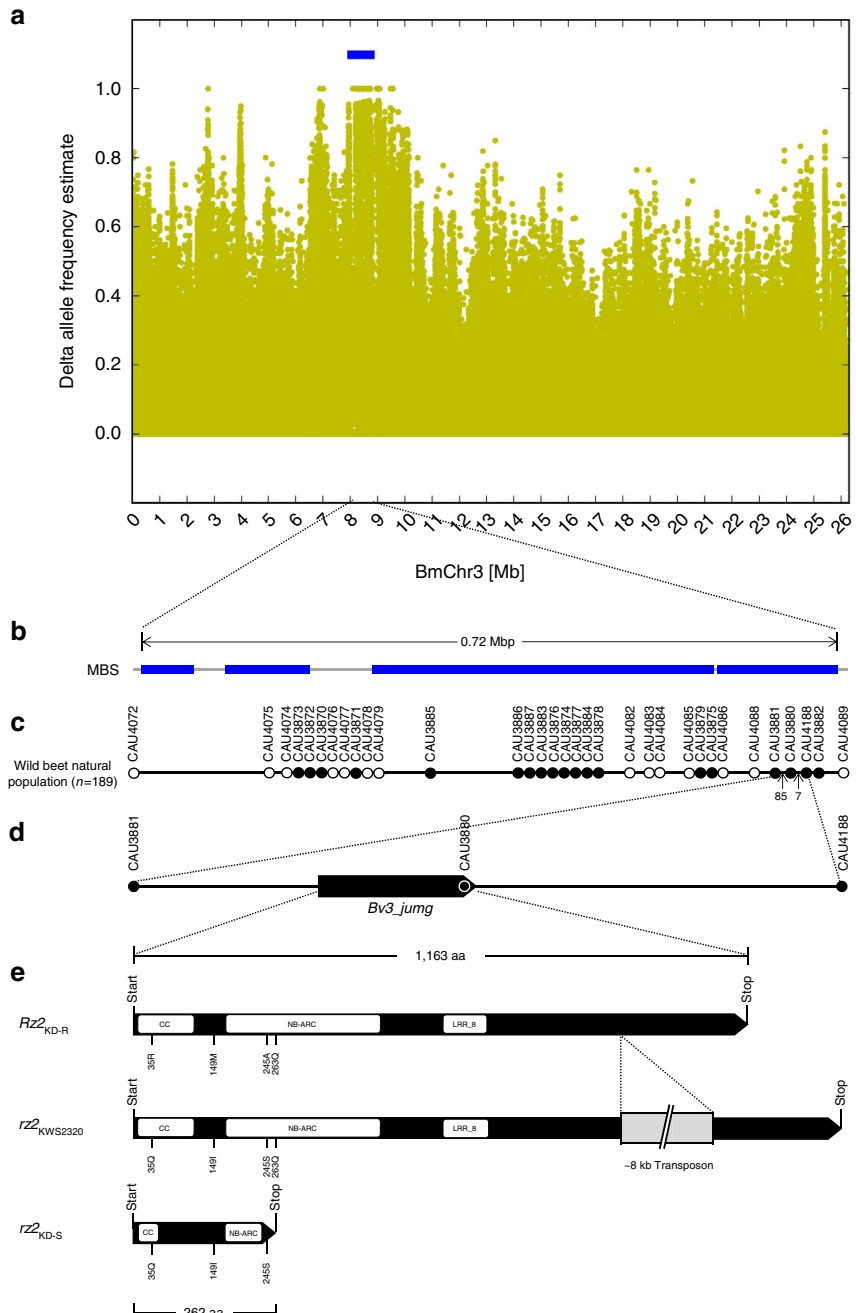

**Figure 2 | The *Rz2* region identified by MBS.** (**a**) Plot of delta-AFe values for BmChr3. The delta-AFe values, indicated by yellow dots, clearly show a skew in the AFe values of the two pools created from four and eight plants unequivocally identified as homozygous resistant and susceptible, respectively. The blue bar indicates the target region defined by genetic intervals from MBS. (**b**) Sketch of the 0.72 Mbp target region. Blue bars indicate genetic intervals identified by MBS. (**c**) Association analysis results after testing 189 families of the Kalundborg population. Black and white circles represent markers associated and not associated with resistance, respectively. Arrows indicate the number of recombinant plants identified. (**d**) Location of *Bv3_jumg* identified as *Rz2* by recombinant analysis. (**e**) Structure of the *Rz2* gene. *Rz2*<sub>KD-R</sub>: allele identified in resistant plants in Kalundborg, *rz2*<sub>KWS2320</sub>: allele present in the susceptible sugar beet genotype KWS2320; *rz2*<sub>KD-S</sub>: allele identified in susceptible plants in Kalundborg. White box: location of conserved domains in the encoded protein sequence for three main domains of a CC-NB-LRR protein; grey box: transposon insertion location; number and letters: predicted amino acid changes.

(ref. 20) and 5,457 (ref. 21) F2 plants were required for identifying sugar beet bolting time genes in a map-based cloning approach. In contrast to multi-parent advanced generation inter-crossed populations that required a minimum of eight crop generations[2], it took only one step of seed increase to obtain a CWR-derived mapping population. Potentially, this step could even be skipped if the trait has high single plant heritability, such as bolting[20,21] or other monogenic traits[22]. Although no direct comparison within

*B. vulgaris* is available, the advantage of CWR populations over populations that are derived from germplasm collections or elite material[13,23,24] is the high population admixture.

The confirmation of the correct identification of the candidate gene *Bv3_jumg* as *Rz2* might have the complication that potentially existing additional sequence-related genes could be off-targets of the RNAi construct. However, read coverage analyses of the *Bv3_jumg* region as well as the correlation of

the inactivated alleles (rz2) with susceptibility argue strongly against the existence of such genes in the B. vulgaris genome. Considering also the marker CAU3880 that is located within the candidate gene, we are confident that Bv3_jung is Rz2.

Our approach relies on naturally occurring variation and is, therefore, independent of the production of a mutant population, unlike the recently published MutRenSeq method[25]. MutRenSeq combines chemical mutagenesis with targeted exome capture sequencing of resistance gene analogues (RGAs). The two approaches are distinct and complementary for the molecular identification of causal resistance genes. However, our approach also allows access to any other type of genes causal for a clear phenotype.

In the work presented, a CWR natural population was explored by application of MBS technology and allowed direct identification of a major locus of high agronomic importance. This was facilitated by the allogamous flowering biology of B. vulgaris ssp. maritima that led to an accumulation of recombination events over generations of reproduction. The process can be streamlined to fit into 1 year, including material development, phenotyping, sequencing and data evaluation given that sufficient personnel is available. Likewise, our approach could also be applied efficiently to other crops with allogamous CWRs, such as rye, sunflower and tomato[26], highlighting the importance of protection and conservation of CWR[27] in their natural habitats.

## Methods

**Natural population sampling and other plant material.** Our study was based on a natural wild beet population located in Kalundborg, Denmark, that was reported to segregate for resistance to rhizomania[6]. From this CWR population, the rhizomania-resistant genebank accessions WB41 and WB42 had been collected and each had been used for mapping a major rhizomania-resistance quantitative trait locus in biparental crosses. The quantitative trait loci had been named Rz2 (WB42) and Rz3 (WB41) and had been mapped to the same genomic region. The mapping data also indicated that the Kalundborg population segregates for only one resistance locus. Therefore, we assumed Rz2 and Rz3 to be identical, referring to them as Rz2 (refs 6,7,28,29).

Seeds were sampled in the coastal area of Kalundborg from 423 open pollinated wild beets and used for one seed increase. Single wild beet plants were grown each in isolation with plants of a cytoplasmatic male sterile sugar beet inbred line that was susceptible to rhizomania. Full sib family seed was harvested from the cytoplasmatic male sterile line plants, separately from the wild beet individuals that provided pollen. Seed production was carried out by the sugar beet breeding companies KWS SAAT SE, Syngenta Agro GmbH and Strube Research GmbH & Co. KG. A total of 189 full sib families with sufficient seed quantity and quality were selected for phenotyping, each derived from a different wild beet individual (Supplementary Data 1). About 40 offsprings were evaluated for each of the families to deduce the genotype of the respective Kalundborg parent individuals.

To obtain an independent population, 382 wild beets of a second CWR population were sampled in the coastal area of Brighton, France. This population has also been identified as a source for rhizomania resistance. Seed was increased as described above for the Kalundborg population by the sugar beet breeding company SESVanderHave N.V., and 129 full sib families derived from the Brighton population with sufficient seed quantity and quality were selected for resistance phenotyping (Supplementary Data 3).

For validation of the association of molecular markers in the Rz2 genomic region with resistance, a panel of 279 sugar beet inbred lines was used. This panel represented the breeding germplasm of KWS, Syngenta and Strube Research. The material covered rhizomania-susceptible and -resistant inbred lines that were phenotypically characterized for the status of Rz1 (ref. 6) and Rz2.

**Rhizomania-resistance phenotyping.** The full sib families derived from the Kalundborg population were phenotyped as follows: 10 single plants from each of 189 full sib families were evaluated by enzyme-linked immunosorbent assay (ELISA). Each ELISA plate held two samples of sap from healthy plants, plus samples for creating a standard curve. The standard curve was based on a series of dilutions: 400, 200, 100, 50, 25, 12, 6, 3, and 1 and, as the end point, a virus-free standard of 0 ng ml$^{-1}$ was used. The 400-ng ml$^{-1}$ standard was made from the sap of sugar beet roots with a high titre of BNYVV. The absolute virus content in that sample was determined using ELISA by comparing it to a purified BNYVV preparation control. Purified BNYVV and ELISA kits were bought from Adgen Ltd. in Scotland. The ELISA protocol described by the manufacturer was followed[11]. Phenotyping was carried out in four independent tests, with two tests using B-type and two tests using P-type BNYVV-containing soil[30,31]. The results

are summarized in Supplementary Data 1. As controls, 100 plants of each of three different sugar beet lines were used in each test: an Rz2 line, an Rz1 line, and one rhizomania-susceptible line.

In addition, 129 full sib families developed from the Brighton population were phenotyped, as described above, in one test using B-type BVYVV[30,31]. As controls, 20 plants of an Rz2 line, an Rz1 line and one rhizomania-susceptible line were used in each test. The results are summarized in Supplementary Data 3.

**Molecular marker analysis.** Leaf samples were taken from each wild beet plant that was grown for production of full sib families. Extraction of genomic DNA was performed using the standard CTAB method[32]. For fine mapping, a total of 33 SNPs distributed within the target region identified by MBS of 0.72 Mbp in length were converted to Kompetitive Allele-Specific PCR (see Supplementary Table 6) or cleaved amplified polymorphic sequence markers and used to genotype the complete panel of 189 families. Kompetitive Allele-Specific PCR genotyping assays were performed in a total volume of 5 μl containing 0.1 ng of genomic DNA according to the manufacturer's guidelines (LGC Genomics LLC, Beverly, MA, USA). Cleaved amplified polymorphic sequence genotyping was used for CAU4188 (WB42 allele: A; alternative allele: C; position in BmChr3: 8,694,501). An amplicon of 457 bp was amplified using the primers 5′-TGGAAGATTGTGCTGAGGA-GA-3′ and 5′-GGACCTTCAGATGGCTTTGC-3′ by PCR. After PCR, an enzyme digestion using SspI was performed according to the manufacturer's guidelines (ThermoFisher Scientific, Southampton, UK). Markers CAU4220, CAU4021 and CAU4022 (Supplementary Table 5) were evaluated by direct Sanger sequencing of amplicons generated by PCR.

For the markers in the 0.72 Mbp target region, deviations from the HWE expectations were estimated using the marker results listed in Supplementary Data 1. The P values for deviation from HWE are listed in Supplementary Table 1.

**Genetic association analysis.** Genotyping data from SNP markers targeting the 0.72 Mbp region on B. vulgaris ssp. maritima chromosome 3 (BmChr3) were used in the analysis. Association analysis for rhizomania resistance was performed in accordance with a Generalized Linear Model method using the TASSEL software (version 3.0). For assuming an association, an adjusted P value (Bonferroni correction) of <0.0015 was required, and $r^2$ (correlation coefficient $r^2 > 0.1$) was used to evaluate the magnitude of the marker effects. The results are summarized in Supplementary Table 7.

**Generation of a WB42 draft genome sequence.** A genome sequence of the Rz2 donor line WB42 was generated. DNA from a single plant was sequenced with the Illumina technology. Data generated from one paired-end (PE) library with an insert size of 600 bp and from two mate-pair (MP) libraries with span sizes of 2.5 and 5 kb, respectively, were combined. PE sequencing on an Illumina HiSeq-2000 instrument was performed using a 2 × 100 cycle protocol for the PE library and a 2 × 50 protocol for the MP libraries. Illumina sequencing reads were quality filtered and trimmed prior to assembly. Reads were removed if they did not pass the chastity filter, had at least one uncalled base, contained at least six bases of the Illumina adaptor sequence in both ends or had less than two-thirds of the bases with quality value $Q \geq 30$ within the first half of the read. Additionally, low-quality read ends marked as 'B' in the quality string ('B-tails') were trimmed off,[14,33] resulting in 390 million high-quality PE read-pairs equivalent to 100-fold coverage of the beet genome. Quality filtering, trimming and redundancy elimination resulted in 30 million MPs for the 2.5 kb span size library and 35 million MPs with span size of 5 kb. Sequence read data were submitted to Sequence Read Archive (SRA). The reads were assembled using SOAPdenovo V1.05 (ref. 34) with standard parameters and a kmer-size of 49. Gaps were filled using SOAP GapCloser v1.12. The final assembly size was 531,940,822 bp in 57,361 scaffolds and contigs >500 bp. The largest scaffold had a length of 899,438 bp, and N50 size was 59,342 bp (see Supplementary Table 3). We refer to this initial assembly as WB42-v0 and to the 57,361 sequences in this assembly as 'WB42-v0 contigs'.

**Concatenation of contigs to yield WB42 pseudochromosomes.** For MBS, we generated WB42 pseudochromosomes from the WB42-v0 assembly. Contigs were ordered according to the sugar beet reference genome sequence RefBeet-1.2 (ref. 14) assuming strong synteny between the two subspecies B. vulgaris ssp. vulgaris and B. vulgaris ssp. maritima. We used BLAST+ (ref. 35) to determine sequence similarity of the WB42-v0 contigs to RefBeet-1.2. The WB42-v0 contigs were sorted according to their RefBeet-1.2 mapping position as follows: (i) the WB42-v0 contigs were aligned to the reference with the dust filter turned off, an e-value cutoff of 1e-50 and a culling limit of 1; (ii) high scoring pairs were combined and the best overall alignment for each WB42-v0 contig was kept; (iii) WB42-v0 contigs were ordered by their mapping position; hits in minus orientation were included as reverse-complement, (iv) the contigs mapping to each chromosome were concatenated to pseudochromosomes with stretches of 50 N's between contigs. We refer to this initial sorted and concatenated assembly as WB42-v1.

**Mapping-by-sequencing.** MBS was performed by applying a workflow[15] that has been modified to fit the CWR setting, initially using WB42-v1 as a target genome

sequence and an adapted interval detection (see below). MBS for *Rz2* was based on the determination of allele frequencies in two DNA pools that were built from four resistant (R1/B2444) and eight susceptible (S1/B2446) individuals. Equal amounts of genomic DNA isolated from freeze-dried leaf samples[32] of single plants were pooled according to the resistance phenotype and sequenced using the Illumina technology.

**Library preparation and sequencing of pooled DNA.** Library preparation for the DNA pools R1/B2444 and S1/B2446 was performed according to the Illumina TruSeq DNA Sample Preparation v2 Guide. DNA from each pool was fragmented by ultrasound shearing. After end repair and A-tailing, individual indexed PE adaptors were ligated to the DNA fragments, which allow a multiplexed PE sequencing run. The adaptor-ligated fragments were size selected on a 2% low melt agarose gel to a size of 350–650 bp. After enrichment, PCR of fragments of the final libraries that carry adaptors on both ends were quantified with a Qubit. The average fragment size of each library was determined on a BioAnalyzer High Sensitivity DNA chip. The samples were sequenced on a HiSeq-2000. Cluster generation for a high output run was done on a cBot using the TruSeq PE Cluster Kit v3, and 2 × 101 bp reads were generated using the TruSeq SBS Kit v3. Sequence read data of both pools were submitted to SRA. After completion of the sequencing runs, basecalling, demultiplexing and fastq file generation was performed using the CASAVA-1.8.2 programs. The results are summarized in Supplementary Table 2.

**Targeted reordering of the WB42 sequence assembly.** The allele frequency values (AFe) determined from sequence data (see below) of the two DNA pools indicated several genetic intervals with highly significant separation between the resistant (R1/B2444) and the susceptible (S1/B2446) pool. When using the WB42-v1 assembly as genome sequence, these intervals were located mainly on BmChr3 but in addition also on four other pseudochromosomes as well as in the unassigned assembly fraction. We concluded that the region of interest was assembled incorrectly and that the WB42-v1 assembly had insufficient quality for MBS and thus set out to improve the WB42 assembly to obtain a better mapping resolution. We focussed on BmChr3 for targeted reordering based on data from genetic markers that place *Rz2* on BmChr3. We manually inspected scaffolds and contigs from RefBeet-1.2 that caused localization of WB42-v0 contigs to BmChr3 and analysed WB42-v0 contigs that might be incorrectly localized to other parts of the genome although the correct localization might be BmChr3.

We used data from a genotyping-by-sequencing experiment[15] to identify true SNPs in regions of BvChr3 that displayed very low variation between the parents of the sugar beet mapping population that was used to build RefBeet-1.2 (ref. 14) and also data from BeetMap-3 (ref. 36). The regions that displayed very low variation between the parents turned out to contribute significantly to the wrong localization of contigs in WB42-v1. The respective RefBeet-1.2 scaffolds were addressed by marker-based genetic anchoring. Validated SNPs were genotyped by direct Sanger sequencing and manual SNP calling from the tracefiles in the KWS1 population[36] and genetically anchored between 62.3 and 68.1 cM on BvChr3 (cM values according to BeetMap-3). Some sections within RefBeet-1.2 scaffolds were identified to be potentially misplaced based on sudden changes of AFe values in the corresponding WB42 scaffolds. To further verify the improved scaffold order on BvChr3, fosmid end sequences[14,37] were evaluated, resulting in support for the proposed scaffold order. Additionally, sections of WB42-v0 scaffolds that (i) displayed a mean allele frequency in the R1 pool differing significantly from 0 (cutoff > 0.05), (ii) corresponded to exactly one WB42-v0 contig and (iii) had a better BLAST hit elsewhere in the *B. vulgaris* reference sequence RefBeet-1.2 were moved to the position indicated by the best BLAST hit. The refined WB42 genome sequence assembly was designated WB42-v2.

**Calculation of allele frequencies from sequencing data.** All operations from postprocessing and mapping of reads over InDel realignment, base quality score recalibration to variant calling and calculation of AFe values were performed on the compute cluster of the CeBiTec. The reads of the sequenced pools R1/B2444 and S1/B2446 were processed by adapter trimming, quality filtered by removing reads with stretches of four consecutive bases with a mean quality value < 30 and removal of bases at the read heads and tails with quality values < 25 (ref. 15). The remaining reads were mapped to the ordered and concatenated WB42 sequences v1 and v2 using BWA-MEM. SNPs and InDels were called using GATK[38]. AFe and delta-AFe values were calculated from the proportion of reads for the two states of a given sequence variant within and between pools, respectively[15].

**Interval identification and prediction of causal variations.** For visual inspection of the results, the delta-AFe values of detected variants were plotted along all WB42-v2 pseudochromosomes or for selected regions (see Fig. 2a and Supplementary Figs 1 and 2). To reduce background noise, only variants with a read coverage between 0.75 and 2.5 times the average coverage of uniquely mapped reads were plotted. Additionally, an established algorithm[15] for automatic interval detection was adapted to the CWR scenario. An interval was defined as a genomic region containing a follow-up of variants with AFe values of ≤ 0.1 in the susceptible (S1/B2444) pool. Interruptions of the follow-up were ignored if they consist of at most 1 variation with an AFe > 0.1, flanked on each side by a variant

with AFe values of ≤ 0.1. Interval detection was started at seed variants with a delta-AFe value close to 1. The exact, lowest delta-AFe value of valid seeds ($X$) was calculated depending on the sequencing error rate and coverage to tolerate a small number of non-supportive reads. $X = (P - 100 \times E/C)$ with phenotypic difference of the pools $P = 0.9$, the coverage of both pools combined for the variant $C$ and an estimate for errors introduced by HiSeq sequencing $E = 0.01$. Adjacent intervals were merged into one if the gap between them was < 1% of the length of the resulting interval after the merge. Finally, intervals < 10 kbp were discarded. The resulting eight intervals detected in WB42-v2 were designated intervals 1–8 and are listed in Supplementary Table 5. Intervals 1–4 were located on BmChr3 and are shown in Fig. 2b and Supplementary Fig. 2.

**Generation of RNA-Seq data from rhizomania infected beets.** Selfing progenies of KD_091343, a rhizomania-resistant plant line derived from WB42, were grown in infected soil using the B-type BNYVV. Seeds were sown in sterile sand. Seven days after sowing, the seedlings were transplanted to 0.25-litre containers filled with a mixture of rhizomania-infected soil and sterile sand in 1:1 proportions. Before transplantation, the soil in the containers was thoroughly soaked with nutrient solution[11]. Root samples of KD_091343 were collected 3, 6, 9, 12, 15 and 20 days after transfer to the infected soil. The infection efficiency of the soil was confirmed by assaying plants of the rhizomania-susceptible line KD_091115, which were grown in parallel.

Total RNA was extracted from the infected root tissue using the peqGOLD Plant RNA Kit (PEQLAB) according to the manufacturer's instructions. After on-column DNase treatment with the peqGOLD DNase I Digest Kit, the RNA was quantified. RNA-Seq (2 × 100 bp) was performed with TrueSeq technology on a HiSeq-1500 by using the Illumina RNA Sequencing Kit complemented with reverse transcriptase according to the instructions of the manufacturer. One barcoded library was created for each of the six time points. Raw RNA-Seq reads were processed as described[39] and mapped to the WB42-v2 genome sequence (see below). In total, RNA-Seq yielded 177 million read pairs (335 million reads) with an average length of the single reads after trimming of 86 bp. Sequence read data of the six libraries were submitted to SRA. For expression detection and gene structure validation, the resulting BAM files were analysed with ReadXplorer2 (ref. 40).

**WB42 ORF prediction.** All scripts were provided by AUGUSTUS[41] and used following AUGUSTUS' instructions. *Ab initio* gene prediction on the WB42-v2 genome sequence was carried out using AUGUSTUS version 2.7 with the optimized sugar beet parameter set[16] and extrinsic hints. The options to predict and print untranslated regions were switched on, the gene model was set to complete and no in-frame stop codons were allowed. Hint files were created based on RNA-Seq data derived from infected root material of KD_091343 (see above). Trimmed paired reads were used for a Tophat (v2.1.0)[42] mapping allowing a maximum fragment length for valid PE alignments of 5,000. The BAM files were quality filtered to minimum mapping quality of 20. For merging and sorting, SAMtools[43] were used. The resulting single BAM file was used as input for *bam2hints.pl* provided by AUGUSTUS to create intron part hints (option --I). To create exon part hints, BAM files were converted to WIG files using *bam2wig* and used to generate the hint file using *wig2hints.pl*. The structural annotation and information on transcript evidence of the 53,190 predicted WB42 genes is available in the file WB42_v2.gff3 (see below). Out of the 53,190 predicted WB42 genes, 33,922 displayed at least 1% transcript evidence.

**Functional annotation of WB42 genes in intervals.** The deduced 74 peptide sequences of the predicted WB42 genes located in the 0.72 Mbp target region on BmChr3 (60 with expression evidence and 14 without) containing the four main MBS intervals (see Supplementary Table 4) were analysed to collect functional annotation information for these genes. Note that the termini of the genetic intervals were determined by the position of variants within WB42-v2 scaffolds, which places these termini within scaffolds. As a result, the sum of the length of the scaffolds that overlap the four genetics intervals was longer than the total length of the four genetic intervals adding up to 0.55 Mbp (see Fig. 2b). For functional annotation, a BLASTp analysis of the WB42-v2 peptide sequences was carried out against the non-redundant protein sequence database (NCBI nr protein, update date: 2016/06/01 containing 88,499,796 sequences, *E*-value cutoff limit e-10), and functional annotation as well as the accession nos. of the best BLASTp hit were extracted. Second, the peptide sequences were analysed by BLASTp for their best hit in the set of all genes from RefBeet[14], including those without transcript evidence (protein sequences extracted from RefBeet.unfiltered_genes.1302.gff3, which is available at http://bvseq.molgen.mpg.de/, *E*-value cutoff limit e-40). Third, sequences were used as query in a Blast Like Alignment Tool (BLAT) analysis[44] against all peptide sequences deduced from BeetSet-2 (ref. 16). BLAT hits > 50% query coverage are listed in Supplementary Data 2. BLAT hits with query coverage > 80% and an identity of > 85% were considered as an indication for the detection of true homologues of WB42 genes in BeetSet-2. Finally, we searched for known resistance gene domains in the WB42-v2 peptide sequences of the target region by using HMM models[14] that had been established for sugar beet. A domain was considered as present if it was at least 95% complete with an *E*-value of ≤ 1e-5. We identified two RGAs; both RGA genes are highlighted in Supplementary Data 2.

One of the two, namely, Bv3_073070_zfyr, has a true intact homologue in BeetSet-2 from the susceptible reference genotype KWS2320, whereas only one, Bm_jumg, is unique to WB42. The protein encoded by this WB42 gene, which received its designation in the initial *B. vulgaris* gene prediction[14], contains the complete functional domains of a CC-NB-LRR protein (CC domain: cd14798; NB-ARC domain: pfam00931; LRR_8 domain: pfam13855).

**Sequence analysis of *Rz2* and *rz2* alleles.** For access to sequences of the *Rz2* gene from different sources, we manually assembled the sequences of the gene from NGS read data and annotated the resulting sequences using CLC Main Workbench (version 6.9.1). The sequences of the two alleles from the Kalundborg population (designated $Rz2_{KD-R}$ and $rz2_{KD-S}$, see Fig. 2e) were assembled from the sequence data of R1/B2444 and S1/B2446, respectively. The sequence of the allele $rz2_{KWS2320}$ was taken directly from RefBeet-1.2 and required only re-annotation since the transposon insertion interfered with automatic structural gene prediction (see also Supplementary Data 2). The sequences of the *rz2* alleles from YMoBv, YTiBv, KDHBv and UMSBv were constructed from their respective assembly data downloaded from http://bvseq.molgen.mpg.de/. Results are summarized in Supplementary Table 9. The presence of a premature stop codon or the insertion of a transposon were observed only in rhizomania-susceptible genotypes.

**Verification of gene identification by RNAi.** For a functional proof of *Bv3_jumg*, a resistant standard sugar beet genotype (line 6921_RR) was transformed with a DNA construct encoding a double-stranded hairpin RNA resulting in 6921_RNAi. The dsRNA was aimed at post-transcriptional gene silencing of the resistant *Bv3_jumg* allele. In order to provide a suitable DNA construct, a defined target sequence region of the resistant *Bv3_jumg* allele of 434 base pair length (nt 247–680 of the mRNA) was selected, amplified by PCR and cloned both in sense and antisense direction in the vector pZFN, a modified version of pLHBA, which is suitable for the synthesis of hairpin structures. This vector contains a double CaMV 35S promoter, two multiple cloning sites separated by intronic sequences, an intron from the *Arabidopsis thaliana* gene *AtAAP6* and the *nos* terminator region. Transformation of sugar beet was performed in accordance with the established protocol[45] and by using kanamycin as selection drug. Following a number of selection steps, successful transformation was examined on transgenic shoots via PCR by detection of the *nptII* gene, the *AtAAP6* intron sequence, the two T-DNA border sequences (LB/RB) and the absence of *vir*. Positive shoots were clonally multiplied *in vitro* to 30 shoots in each case, rooted and transferred into soil in the greenhouse. Approximately 2 weeks later, the transgenic sugar beet plants (6921_RNAi) were transplanted into a rhizomania assay as described above. As controls, the non-transformed-resistant genotype (6921_RR) that was used for RNAi transformation and the susceptible line D108_ss were included (Supplementary Fig. 4).

**Data availability.** The Illumina sequence read data were submitted to the NCBI SRA. Reads used to create the WB42-v0 assembly received accession no. SRP078074, and reads of the two DNA pools from the Kalundborg population received accession nos. SRX1922267 (R1/B2444) and SRX1940996 (S1/2446). Reads from the RNA-Seq experiment with infected root tissue received accession no. SRX1924131.

The contig and scaffold sequences that were used to build the WB42-v2 assembly were collected in FASTA format (WB42_v2.fasta). The sorting and concatenation of these sequences into pseudochromosomes for WB42-v2 is described in an AGP file (WB42_v2.agp). The three files WB42_v2.fasta, WB42_v2.agp and WB42_v2.gff3 are available from http://dx.doi.org/10.5447/IPK/2017/3.

The authors declare that all other data supporting the findings of this study are available from the corresponding authors upon request.

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

## Acknowledgements

This work was supported by the German Ministry of Food and Agriculture (BMEL) grant FKZ 2814505310 to F.J.K.-O. and M.V. and the German Ministry of Education and Science (BMBF) grant FKZ 0315957D (Acronym: NuGGET) to B.W. We thank Monika Bruisch for help with plant sampling, as well as Meike Pfeiler and Prisca Viehöver for excellent wet lab work. In addition, we thank Christian Jung for providing laboratory infrastructure and support.

## Author contributions

F.J.K.-O. conceived the study; G.G.C.-G. and F.J.K.-O. designed and analysed the mapping experiments; K.S., O.T., D.B., W.M. and J.C.L. provided the transgenic RNAi results; G.G.C.-G., D.R., D.H., F.J.K.-O. and B.W. wrote the manuscript; D.R., D.H., T.R.S. and B.W. established the pseudochromosomes of WB42 using RefBeet1.2 as backbone, performed the NGS analysis, genome annotation and RNA-Seq analysis; A.W.S., T.K., M.V., H.T., H.U., J.C.L. and W.M. developed the material, did resistance tests and selected elite material for the validation panel; L.F. recommended exploitation of the Kalundborg and Brighton CWR populations; G.G.C.-G., D.H. and S.L.M.F. carried out marker analysis, resistance test analysis, association analysis and sequence analysis of the candidate genes; A.M., J.D., I.G. and H.H. sequenced and assembled the genome of WB42; and M.S. sequenced B2444 and B2446.

## Additional information

**Competing interests:** O.T., D.B., W.M. and J.C.L. have filed a patent application that includes the molecular complementation of *Rz2* (WO 2014/202044 A1). All other authors declare no competing financial interests.

