## [Peer Review File · Nature Communications]

Reviewers' comments:

Reviewer #1 (Remarks to the Author):

Capistrano et al. used mapping by sequencing of phenotypic pools of resistant and susceptible individuals of a natural wild beet population of the Danish coastline to spot a CC-NB-LRR gene as candidate for the rhizomania resistance locus Rz2. The use of a wild beet population provided excellent genetic resolution due to low LD, compared to mapping in a bi-parental mapping population. It is an excellent example how population genomics in crop wild relatives can support efforts of resistance breeding in important crop species. A draft assembly of the Rz2 resistance donor was used for MBS read mapping, which provided a work around for the case that resistant and susceptible beet have too divergent haplotypes at the Rz2 locus. Gene annotation in this draft assembly was performed with RNAseq data obtained from the resistant genotype. A CC-NB-LRR gene, truncated in the susceptible haplotype was identified as candidate gene. A marker derived from this gene was always associated with resistance in an independent CWR population from France and defective alleles of the gene were consistently associated with susceptibility.

This is an excellent example of using state-of-the-art technology in combination with CWR population genomics and should reach broad interest, because it excellently illustrates how advanced the approaches are even in small crops now and how long lasting genetic problems can be elegantly solved if the appropriate strategy and the optimal tools are combined.

The only limitation of the study I see is that the authors do not provide a functional proof for their findings but only the genetic evidence for the identified single candidate gene. An induced Crips/CAS9 knock-out would make this a completely sound study.

Reviewer #2 (Remarks to the Author):

This manuscript describes identification of Rhizomania resistance Rz2 using a natural population in wild beets. Authors used so called a "crop wild relatives (CWR)" population that is growing along the coast of Kalundborg, Denmark, and is known to segregate for Rz2. They performed progeny tests by crossing 189 wild beets with a Rhizomania susceptible sugar beet line and revealed genotypes of each wild beet for Rz2. To do fine mapping of Rz2, authors selected homozygous resistant and susceptible lines and created two bulks consisted with resistant and susceptible beets for mapping-by-sequencing (MBS). The Rz2 donor line WB42 was also sequenced and yielded sequence of a genomic region of 0.72 Mbp (BmChr3) where Rz2 is presumed to be located. In addition, 60 genes in the target region were obtained by transcriptome sequencing of WB42 allowed. Out of 33 SNPs distributed over the 0.72 Mbp target region, 19 markers were found to be associated with resistance in the Kalundborg population. The best association was found for marker CAU3380 within a CC-NB-LRR gene. This association was validated using another CWR population and diversity panels of sugar beet inbred lines from three breeding companies. Sequence analysis of Rz2 alleles revealed allelic variations including a SNP causing stop codon or a transposon insertion. Overall the work presented in this paper delivers very useful

information in terms of developing useful markers for breeding Rhizomania resistance in sugar beets. However, I am concerning whether the work is significant and novel enough to be published in Nature Communications. First of all, even though they claim that they are the first group to use so called a "crop wild relatives (CWR)" population, they already knew the position of the Rz2 gene and just used natural recombination of allogamous plants. Therefore, their method is just another way of a fine mapping approach. I think the population is very unique in this study, and doubt commonality of the CWR population segregating for useful genes in other crops. They claim that they used MBS for the Rz2 gene identification, but is another way of commonly used bulked-segregant analysis. Another concern is validation of the candidate gene for Rz2. They should confirm and characterize the function of the candidate gene for Rz2 some other way including transgenic analysis, qRT-PCR, protein-protein interaction, localization of the gene etc.

Reviewer #3 (Remarks to the Author):

Capistrano et al (Kopisch-Obuch) set out to fine map and identify the Rhizomania resistance gene Rz2 previously introgressed into elite sugar beet germplasm from the wild relative *Beta vulgaris* ssp. *maritima*. What sets this study apart from earlier crop disease resistance (R) gene cloning projects is the use of a diversity panel from the wild beet relative to fine map the target R gene by association genetics (i.e. phenotyping and genotyping the panel followed by correlating phenotype with genotype). The authors rightfully point out that previous studies, on the other hand, have relied much more heavily on synthetically structured populations e.g. F2, RIL, MAGIC or mutants, which can be labour intensive, costly, and time-consuming to generate. This study highlights two advantages of a crop wild relative diversity panel, namely (i) not having to make your own structured population (nature has already done it for you!), and (ii) the high degree of historical recombination in wild germplasm whereby less individuals need to be genotyped and phenotyped (i.e. you save money and time!).

While the study is elegant and the manuscript well written, we do, notwithstanding, have some concerns as detailed below.

Major points:

1. Most importantly, by not combining functional studies with their association genetics, the authors do not show unambiguously that they have cloned Rz2. The authors define a candidate region in the wild beet genome by resequencing a bulk resistant pool (4 accessions) and a bulk susceptible pool (8 accessions) and projecting the SNPs onto a de novo wild beet reference genome. The authors identified an enrichment of SNPs from the resistant pool on chromosomes 3, 4, 8 and an unmapped scaffold (Figure S1). It was previously shown that Rz2 maps to chromosome 3, so within the chromosome 3 candidate region the authors then chose 33 SNPs and genotyped a 189-member large wild beet panel to identify historical recombination events to narrow down the candidate region (Figure 2d). Within this region, the authors identify an expressed candidate CC-NBS-LRR gene. This gene was found to be mutated (early stop codon) or containing a transposon insertion in two susceptible individuals (Figure 2e).

While this constitutes strong genetic evidence that the identified gene is Rz2, other possibilities cannot be formally excluded. For example:

(i) It is well known that R genes belong to large multi-gene families. Given the high sequence identity often present between related members (paralogues) at the same locus it is notoriously difficult to assemble R gene loci from short-read data. In such a scenario, another paralogue at the Rz2 locus could have collapsed during the whole genome assembly, and this other gene could be the real Rz2.

(ii) Numerous other SNPs with strong association were identified which could not be assigned to a chromosome or which were postulated (but not confirmed) to belong to chromosome 3 (lines 142-146). Therefore, one or more of these SNPs could be the true causative variants.

To overcome the limitations of correlation genetics (whether based on natural or synthetic populations), R gene cloning exercises therefore typically employ functional assays such as (i) mutagenesis to knock out the candidate gene, and/or (ii) transformation to confer resistance to another, previously susceptible cultivar. In the absence of such confirmative experiments, the authors cannot unequivocally state that they have cloned Rz2. In light of this, the authors make some misleading statements, e.g.:

a) In the Title: "Identification of the Rhizomania resistance gene Rz2..."

b) Line 208: "The identification of Rz2..."

We would suggest that the authors transform their Rz2 candidate into a susceptible cultivar and show that this confers resistance, or that they knock out the candidate gene in a resistant cultivar and show that this abrogates resistance. In the absence of these experiments, the authors would, at the very least, have to tone down their statements in the title and text to indicate that they have identified a candidate gene for Rz2.

2. The authors state that crop wild relative diversity panels can "serve to directly identify genes underlying [...] traits" (line 73, and elsewhere) and highlight this as one of the key advantages over other gene identification approaches that rely heavily on synthetic populations. However, the association genetics approach employed in this study identified positive correlations across several regions of the genome. The authors skirt over the inconvenient fact that data derived from bi-parental populations had previously mapped Rz2 to chromosome 3. The way Figure 2 in the main text is presented therefore seems misleading, as it suggests that chromosome 3 was the only region that was identified in the association genetics approach. It would be more honest if Figure S1 were included in the flow-diagram above Figure 1a. On the topic of Figures S1, why were the regions with delta-AFe values of 1 on chromosome 4, 5, 8, 9 and "Random", discarded?

3. Lines 118-120. How can the authors assume a single locus (rather than multiple loci) in the Kalundborg population by crossing to a susceptible line and phenotyping the progeny? Surely genetic mapping of resistance in multiple individuals would be required to assume a single locus. Also, the assumption of a single locus is potentially at odds with the association genetics data presented in Figure S1.

4. The authors point out that one of the advantages of association genetics in their present study is that a small number of lines had to be genotyped and phenotyped ($N < 200$) (lines 208-211). They contrast this to two map-based cloning studies that used biparental populations in which 8,283 (Pin et al) and 5,457 (Dally et al) F2 plants were screened. However, the comparison is not entirely fair because in map-based gene approximation it is common practise to perform a PCR screen on the large population to pre-select

recombinants for phenotyping. This was indeed the case in the Pin et al study in which only 107 recombinants out of the 8,282 F2 plants were phenotyped.

5. The authors mention in their conclusion paragraph (lines 228-230):

“In the work presented, a CWR natural population was explored for the first time by application of MBS technology, allowing direct identification of a major locus of high agronomic importance within one year, including material development.”

This statement grinds against the fact that the crosses described in this manuscript were first reported in an abstract published in 2012: <https://www.google.co.uk/?client=firefox-b-ab#q=beta+world+network+Kopisch-Obuch+rz2+abstract>).

6. The Method reported here appears to be very dependent on the assembly of a high quality reference genome. Lots of manipulation was required, including inspection of Fosmid end sequences, to achieve this – see lines 143 to 173.

7. The authors generated ~29 Gb RNAseq data to support gene annotation, and say that this “supported 100% transcript evidence” for Rz2 (line 155). Could the authors please explain what is meant by “100% transcript evidence”? A figure showing the RNAseq reads relative to the genomic sequence would be welcome, perhaps as Supplementary information. The RNASeq data was collected over a time course. Was there any indication of induction of the Rz2 candidate gene following pathogen challenge?

Minor points

1. Please mention the genome size of sugar beet in main text (near line 130) or in Supplementary information (near line 90) when dealing with the generation of the draft genome assembly.

2. Supplementary Tables 1, 6, and 11 appear to have become truncated during the conversion to Pdf. It would be better to provide these as Excel Tables.

3. Line 169 to 171. Supplementary Table 9 is cited to support the finding that the French wild beet diversity panel identified the marker CAU3880 as associated with Rhizomania resistance. However, this Table shows something different, namely the: “Geographical positions and resistance test results of 129 wild beet plants from the Brighton population”.

4. Supplementary Table 3. We expect that R1/B2444 and S1/B2446 are the resistant and susceptible pools, respectively, but please confirm this in a footnote.

5. In the association analysis for Rhizomania resistance, it seems that p-1 value (p-value \leq 0.01), and not the adjusted p-value (Bonferroni correction), is being used as the criterion for assuming significant association with the phenotype. How was this criterion decided?

Response to reviewers' comments:

We thank all three reviewers for their time and effort which have significantly contributed to improvements of the manuscript.

The line numbers included below refer to the final, EndNote formatted manuscript.

Response to the comments of Reviewer #1:

The only limitation of the study I see is that the authors do not provide a functional proof for their findings but only the genetic evidence for the identified single candidate gene. An induced Crips/CAS9 knock-out would make this a completely sound study.

In the revised version, we have included additional data that were contributed by additional authors (see authors list). These data demonstrate that *Rz2* is the functionally relevant gene, because inactivation of *Rz2* by RNAi in a resistant line/genotype caused this line to become sensitive (new Supplementary Fig. 3).

Response to the comments of Reviewer #2:

*First of all, even though they claim that they are the first group to use so called a "crop wild relatives (CWR)" population, they already knew the position of the *Rz2* gene and just used natural recombination of allogamous plants. Therefore, their method is just another way of a fine mapping approach. I think the population is very unique in this study, and doubt commonality of the CWR population segregating for useful genes in other crops.*

It is correct that prior to this study we knew the approximate position of the *Rz2* gene. However, we used a genome wide mapping approach that did not depend on this *a priori* information about the gene location. Also, the best position estimate before our study was an interval spanning as much as 14 cM (Gidner *et al.*, 2005). In contrast, Pin *et al.* (2010) based their study on an interval as small as 0.37 cM.

We did not claim uniqueness for using natural recombination in fine mapping. Instead, we claim uniqueness for using an allogamous CWR *in situ* population that has undergone repeated (likely more than hundred) generations of recombination within a limited spatial distribution. The combination of these factors provided a mapping population with high population admixture and a resolution on a single gene level despite a population size of less than 200. To our knowledge, this has not been demonstrated in other crops.

We disagree that our populations are unique in a way that our study lacks commonality. First, in addition to the Danish population, we identified a further suitable CWR population from France in which we could confirm our results. Second, there is no evidence that similar CWR populations do not exist for other traits or other crop species. However, as we pointed out in the manuscript, the CWR-based approach should work primarily for allogamous species such as rye, sun flower, and tomato.

They claim that they used MBS for the Rz2 gene identification, but it is another way of commonly used bulked-segregant analysis.

We agree with Reviewer #2 that MBS is a variation of BSA, a fact that is discussed in the paper by Korbinian Schneeberger (Nature Reviews Genetics, 2014; reference No 12). However, usual BSA would surely fail with two pools of 4 and 8 individuals. It is the power of CWR populations that allows to successfully identify the causal gene.

Another concern is validation of the candidate gene for Rz2. They should confirm and characterize the function of the candidate gene for Rz2 some other way including transgenic analysis, qRT-PCR, protein-protein interaction, localization of the gene etc.

As mentioned above in reply to Reviewer #1, we have now included additional data that demonstrate that *Rz2* is the functionally relevant gene. Addressing also the detailed molecular function of *Rz2* would go far beyond the present study.

Response to the comments of Reviewer #3:

1. *Most importantly, by not combining functional studies with their association genetics, the authors do not show unambiguously that they have cloned Rz2.*

As mentioned above in reply to Reviewer #1, we have now included additional data that demonstrate that *Rz2* is the functionally relevant gene.

In such a scenario, another paralogue at the Rz2 locus could have collapsed during the whole genome assembly, and this other gene could be the real Rz2.

We agree with Reviewer #3 that genes that are members of well conserved gene families are sometimes difficult to assemble. However, in the case of *Rz2*, two independent assemblies placed the gene in the same syntenic context. The scaffold containing *Rz2* from the WB42 assembly is fully syntenic to the corresponding region from the reference (KWS2320). In our opinion, this argues clearly against an assembly error that affects *Rz2*. To clarify this point, we added text to the revised manuscript (page 6 lines 156 to 160):

"Apart from its truncation in the reference sequence genotype, Bv3_jumg was located in a region that displays strong synteny between WB42 and KWS2320. The region was well assembled, co-linear, and displayed no indication of read coverage deviation, indicating that no duplicated or paralogous loci from elsewhere in the genome interfered."

2. *The authors state that crop wild relative diversity panels can “serve to directly identify genes underlying [...] traits” (line 73, and elsewhere) and highlight this as one of the key advantages over other gene identification approaches that rely heavily on synthetic populations. However, the association genetics approach employed in this study identified positive correlations across several regions of the genome.*

It is correct that the MBS approach using 4 + 8 plants resulted in some remaining sequences that could not be unequivocally assigned to the target region on BvChr3. However, only the marker CAU3880 which is located within *Rz2* correlated throughout the breeding panel and all genotypes of the Kalundborg population (189 plants). To clarify the point, we have added mapping data to address the three additional loci (four intervals). The new data show that these loci are not co-segregating with the resistance locus. This result excludes these regions as now described in the revised manuscript (page 5, lines 137 to 144). It should be noted that the core problem is the quality of the reference sequence, not MBS as such (see also below).

*The authors skirt over the inconvenient fact that data derived from bi-parental populations had previously mapped *Rz2* to chromosome 3. The way Figure 2 in the main text is presented therefore seems misleading, as it suggests that chromosome 3 was the only region that was identified in the association genetics approach. It would be more honest if Figure S1 were included in the flow-diagram above Figure 1a. On the topic of Figures S1, why were the regions with delta-AFe values of 1 on chromosome 4, 5, 8, 9 and “Random”, discarded?*

We are sorry that the difference between the WB42 assembly v1 and v2 did not become sufficiently clear. The genome regions mentioned by Reviewer #3 were misplaced initially in the assembly WB42-v1, but were moved to their correct location in assembly WB42-v2 as described in the revised manuscript (page 5, lines 129 to 131). The remaining signals visible in

Supplemental Figure 1 outside of Chr3 were caused by assignment errors (not assembly errors), a phenomenon that is well known for re-sequencing approaches that position (assign) contigs from a given genotype according to sequence similarity to a reference sequence. This is explained in the revised version of the manuscript on page 15, lines 308 to 405.

3. *How can the authors assume a single locus (rather than multiple loci) in the Kalundborg population by crossing to a susceptible line and phenotyping the progeny? Surely genetic mapping of resistance in multiple individuals would be required to assume a single locus. Also, the assumption of a single locus is potentially at odds with the association genetics data presented in Figure S1.*

We draw our conclusion of a single major gene from the genetic segregation of 189 full sib families that each represents a different individual of the Kalundborg population. In this regard, we studied multiple individuals as required for supporting a single major locus. Also, each full sib family was represented with a sample size of about 40 individual offspring plants for phenotyping (see improved text on page 9, lines 257-258 of the manuscript). In addition, the marker data (CAU3880 and others) presented in our manuscript clearly confirm: (i) the cited data regarding the localisation of *Rz2* on Chr3, (ii) our initial assumption based on all data generated from the Kalundborg population including HWE data (Supplementary Table 2) that were compatible with a single locus on Chr3, and (iii) the main region detected by MBS containing 60+14 genes (of about 34,000 in total) on Chr3. The potential discrepancy mentioned by Reviewer #3 was resolved by detecting assignment errors (see above) that are known to occur in reference-guided assembly generation.

4. *The authors point out that one of the advantages of association genetics in their present study is that a small number of lines had to be genotyped and phenotyped (N< 200) (lines 208-211). They contrast this to two map-based cloning studies that used biparental populations in which 8,283 (Pin et al) and 5,457 (Dally et al) F2 plants were screened. However, the comparison is not entirely fair because in map-based gene approximation it is common practise to perform a PCR screen on the large population to pre-select recombinants for phenotyping. This was indeed the case in the Pin et al study in which only 107 recombinants out of the 8,282 F2 plants were phenotyped.*

It is correct that through gene approximation the amount of phenotyping can be limited. However, this requires previous information on the location of the target gene. The study of Pin et al. (2010) was based on a fine mapping study of El-Mazawy et al. (2002) where in a

population of 2,134 individuals two flanking markers of the target gene were mapped at distances of 0.14 and 0.23 cM. Still, Pin *et al.* phenotyped their complete biparental mapping populations with more than 8,000 individuals but limited only further marker analysis to 107 recombinants. In contrast, our CWR based approach does not require such previous mapping information, although it might be helpful when it exists.

5. *The authors mention in their conclusion paragraph (lines 228-230):*

"In the work presented, a CWR natural population was explored for the first time by application of MBS technology, allowing direct identification of a major locus of high agronomic importance within one year, including material development."

This statement grinds against the fact that the crosses described in this manuscript were first reported in an abstract published in 2012: <https://www.google.co.uk/?client=firefox-b-ab#q=beta+world+network+Kopisch-Obuch+rz2+abstract>).

It is not clear to us what Reviewer #3 wants to suggest. It is common practice that project progress is exchanged with colleagues at meetings, and this information exchange prior to publication is of central importance to the scientific community as a whole.

With regard to timing: even if in our case the process took longer because of various aspects of research at a university, it was possible to carry out the work in about 12 months counted as time without gaps caused by external factors. Anyway, we have rephrased the sentence. It now reads (page 8, lines 229 to 233):

"In the work presented, a CWR natural population was explored by application of MBS technology and allowed direct identification of a major locus of high agronomic importance. The process can be streamlined to fit into one year, including material development, phenotyping, sequencing, and data evaluation given that sufficient personnel is available."

6. *The Method reported here appears to be very dependent on the assembly of a high quality reference genome. Lots of manipulation was required, including inspection of Fosmid end sequences, to achieve this – see lines 143 to 173.*

Yes, we can confirm that sequenced-based mapping is dependent not only on a high quality reference sequence, but also on re-sequencing data of the relevant genotype unless the studied genotype is closely related to the genotype used to generate the reference genome sequence of the species studied. However, given that the sugar beet reference genome sequence was assembled with 454 sequencing data as a backbone, and that new third generation DNA sequencing technology produces very much improved genome sequences (see e.g. VanBuren *et al.* (2015), *Nature* 527:508-511. Single-molecule sequencing of the desiccation-tolerant grass

Oropetium thomaeum), this will not be a significant problem in the future. It is unfortunate that the fact mentioned by Reviewer #3 does not cause funding agencies to fund the improvement of existing reference genome sequences to "state of the art". One argument is that targeted improvement is possible, and this is the route we followed for *Rz2*.

7. *The authors generated ~29 Gb RNAseq data to support gene annotation, and say that this "supported 100% transcript evidence" for Rz2 (line 155). Could the authors please explain what is meant by "100% transcript evidence"? A figure showing the RNAseq reads relative to the genomic sequence would be welcome, perhaps as Supplementary information. The RNASeq data was collected over a time course. Was there any indication of induction of the Rz2 candidate gene following pathogen challenge?*

The classification according to transcript evidence is a parameter provided by the AUGUSTUS gene annotation package if *ab initio* gene calling is combined with interpretation of mapped RNA-Seq data. Basically, it describes the support of the gene model produced by AUGUSTUS from the RNA-Seq data incorporated for structural gene calling.

Since *Rz2* is a single exon gene, the RNA-Seq read mapping is not very informative. For this reason, we did not include such a figure. Finally, apart the fact that *Rz2* is expressed following pathogen challenge, we have not observed induced gene activity in the RNA-Seq dataset produced.

Minor points

1. *Please mention the genome size of sugar beet in main text (near line 130) or in Supplementary information (near line 90) when dealing with the generation of the draft genome assembly.*

Thanks for the hint! We have included the missing information on page 5, lanes 129 to 131: "RefBeet1.2 covers 567 Mbp of the *Beta vulgaris* ssp. *vulgaris* genome that has an estimated haploid genome size of about 730 Mbp in 9 chromosomes."

2. *Supplementary Tables 1, 6, and 11 appear to have become truncated during the conversion to PDF. It would be better to provide these as Excel Tables.*

We provided these tables initially as .xlsx files, but they were automatically converted during the upload process. We will try to avoid this error when re-submitting.

3. *Line 169 to 171. Supplementary Table 9 is cited to support the finding that the French wild beet diversity panel identified the marker CAU3880 as associated with Rhizomania resistance. However, this Table shows something different, namely the: "Geographical positions and resistance test results of 129 wild beet plants from the Brighton population".*

Sorry for this incomplete connection. Since only marker CAU3880 showed association with rhizomania resistance, we included the result in the text. We have rephrased the sentence on page 6 (lines 168 to 171):

"Notably, CAU3880 was also the only marker that displayed association with rhizomania resistance (P value = $6.1E-4$, $r^2 = 0.17$) in an independent second CWR population that was sampled in Brighton, France (see Supplementary Table 10)."

4. *Supplementary Table 3. We expect that R1/B2444 and S1/B2446 are the resistant and susceptible pools, respectively, but please confirm this in a footnote.*

Thanks for the hint. Corrected in Supplementary Table 3.

5. *In the association analysis for Rhizomania resistance, it seems that p-1 value (p-value ≤ 0.01), and not the adjusted p-value (Bonferroni correction), is being used as the criterion for assuming significant association with the phenotype. How was this criterion decided?*

The Bonferroni correction was used, considering all 33 tested markers. We have rephrased this sentence, including the adjusted P value after Bonferroni correction (page 11, lines 309 to 311):

"For assuming an association, an adjusted P value (Bonferroni correction) of less than 0.0015 was required, and r square (correlation coefficient $r^2 > 0.1$) was used to evaluate the magnitude of the marker effects."

REVIEWERS' COMMENTS:

Reviewer #1 (Remarks to the Author):

The authors have addressed satisfyingly my concerns.

Reviewer #3 (Remarks to the Author):

Reviewer 3: Comments on revised manuscript (NCOMMS-16-18299A)

In their revised manuscript, Weisshaar and colleagues have now responded to most of the concerns I raised. Regarding the functional analysis of Rz2 by RNAi it would have been more compelling had Rz2-mediated resistance been knocked out by CRISPRs (as suggested by one reviewer) and/or conferred to a susceptible beet line by transformation, given (i) the well-known potential for RNAi to silence multiple sequence-related targets, combined with (ii) the observation that R genes typically occur in clusters of genes with high degrees of sequence conservation. As a buffer against this potential weakness, the authors state in their revised manuscript that they did not observe "read coverage deviation" at Rz2 indicative of Rz2 not suffering from a collapsed assembly (indicative of multiple homologues). It would certainly be advisable to show this data in a supplementary figure.

There are therefore still some 'chinks in the armour' and alternative scenarios concerning the nature of Rz2 cannot be formally excluded. I am, on balance, however, satisfied that most likely the correct gene for Rz2 has been identified. Taking this into consideration and given the statements addressing my other concerns, I do not have any major objections to the claims made in the revised manuscript.

Response to the comments of Reviewer #3:

1) Regarding the functional analysis of Rz2 by RNAi it would have been more compelling had Rz2-mediated resistance been knocked out by CRISPRs (as suggested by one reviewer) and/or conferred to a susceptible beet line by transformation, given (i) the well-known potential for RNAi to silence multiple sequence-related targets,

Following the advice of the editor, we have discussed this potential weakness of RNAi in the Discussion section of the manuscript.

2) combined with (ii) the observation that R genes typically occur in clusters of genes with high degrees of sequence conservation. As a buffer against this potential weakness, the authors state in their revised manuscript that they did not observe "read coverage deviation" at Rz2 indicative of Rz2 not suffering from a collapsed assembly (indicative of multiple homologues). It would certainly be advisable to show this data in a supplementary figure.

Thanks for this hint - we have included a new "Supplementary Fig. 3" that displays a read coverage plot of the Rz2 locus.